# STABILIZING POLICY GRADIENTS FOR STOCHASTIC DIFFERENTIAL EQUATIONS BY ENFORCING CONSISTENCY WITH PERTURBATION PROCESS

## ABSTRACT

Deep neural networks parameterized stochastic differential equations (SDEs) received increasing attention from the machine learning community due to their high expressiveness and solid theoretical foundations, with a wide range of applications in generative models. However, maximizing likelihood of training data, the objective of generative models, does not always meet our requirements in many real-world problems. Fortunately, introducing reinforcement learning (e.g., policy gradient) here to maximize a reward, using SDE-based policy, may bridge this gap. Nevertheless, when applying policy gradients to SDEs, since the policy gradient is estimated on a finite set of trajectories, it can be ill-defined, and the policy behavior in data-scarce regions may be uncontrolled. This challenge compromises the stability of policy gradients and negatively impact sample complexity. To address these issues, we propose constraining the SDE to be consistent with its associated perturbation process. Since the perturbation process covers the entire space and is easy to sample, we can mitigate the aforementioned problems. Our framework offers a general approach for training SDEs using policy gradients, allowing for a versatile selection of policy gradients to effectively and efficiently train SDEs. We evaluate our algorithm on the task of structure-based drug design and optimize the binding affinity of generated ligand molecules. Our method achieves the best Vina score $(-9.07)$ on the CrossDocked2020 dataset.

## 1 INTRODUCTION

Deep neural networks parameterized stochastic differential equations (SDEs) have garnered significant interest within the machine learning community, owing to their robust theoretical underpinnings and exceptional expressiveness. In recent years, SDEs have taken center stage in generative modeling (Song & Ermon, 2019; Ho et al., 2020; Song et al., 2021b), with a myriad of applications ranging from image generation to molecular generation, and beyond (see Yang et al. (2022) for a comprehensive survey). SDE generates samples by simulating the following process from time 1 to 0:

$$dx_t = \pi_\theta(x_t, t)dt + g(t)d\bar{\omega} \tag{1}$$

where $x_0$ is the generated sample, $x_1$ is typically drawn from $\mathcal{N}(0, I)$, $\pi_\theta$ is the model, $\bar{\omega}$ is the reverse Wiener process, and $g(t)$ is a scalar function of time and known as diffusion coefficient. Generative modeling aims to approximate the data distribution and, therefore, trains SDEs to maximize the likelihood. However, in many real-world applications, the objective is to maximize a reward. For example, in structure-based drug design (Anderson, 2003), our objective is usually to generate molecules with some desired properties, such as high binding affinity (Du et al., 2016). In these cases, it is natural to train SDEs with reinforcement learning (RL) (Sutton & Barto, 2018; Black et al., 2023; Fan et al., 2023).

RL focuses on learning a policy that maximizes rewards through interactions with a given environment. The foremost class of policy optimization algorithms in RL is the policy gradient method (Sutton & Barto, 2018), which refines the policy network by following the gradient of expected rewards with respect to the policy network parameters. Policy gradient algorithms have demonstrated efficiency and effectiveness in a wide range of applications, such as control systems, robotics, natural language processing, and game playing, among others.

The SDE in Eq. 1 can be interpreted as a Markov Decision Process (MDP) (Black et al., 2023). Therefore, we can directly apply policy gradient to train SDEs. However, in practice, policy gradients are typically estimated using a limited set of sampled trajectories, which can lead to two practical issues (please see Sec. 3 for detailed discussion.): (1) Insufficient data in the vicinity of a trajectory can result in an inaccurate estimation of the policy gradient. This ill-defined policy gradient may cause instability during the training process; and (2) When the simulation of the SDE in Eq. 1 encounters data-scarce regions, the policy in these regions may not be well-trained, leading to uncontrolled behavior and less efficient interactions with the environment. These challenges have hindered the application of policy gradients to SDEs.

**Contribution**: To address the aforementioned challenge, we present a novel framework for training SDEs using policy gradient methods. The main idea behind our method is to enforce the SDE in Eq. 1 to be consistent with its associated perturbation process (See Defn 1 for the definition.). Given that the perturbation process covers a wide space and is easy to sample, we can resolve the aforementioned challenge, and further estimate the policy gradient in a more stable and efficient way. This consistency can be easily maintained using techniques derived from diffusion models. Moreover, we introduce an innovative actor-critic policy gradient algorithm tailored for consistent SDEs. We empirically validate our algorithm on structure-based drug design (SBDD), and optimizes the binding affinity, one primary objective in SBDD, of generated molecules. Our algorithm achieves state-of-the-art Vina scores ($-9.07$) on the CrossDocked2020 dataset (Francoeur et al., 2020).

## 2 Preliminary

We discuss preliminary information in this section including reinforcement learning, SDE-based generative models, and the approach to modeling SDEs as a Markov Decision Process.

### 2.1 Reinforcement learning and policy gradients

Reinforcement learning uses the Markov Decision Process (MDP) to model the decision-making process. An MDP is a tuple $(S, A, \mathcal{T}, R, \tilde{P})$ where $S$ is the state space, $A$ is the action space, $\mathcal{T} : S \times A \to \Delta(S)$ is a transition kernel where $\Delta(S)$ denote the set of distributions over $S$, $R : S \times A \to \mathbb{R}$ is a reward function and $\tilde{P}$ is the initial state distribution. Without loss of generality, we assume the MDP terminates after $N$ steps.

The RL agent takes actions sampled from a policy $\pi$, which is a mapping from $S \times [N]$ to the distribution over $A$, where $[N] := 1, \ldots, N$. The reward of $\pi$ is defined as $R(\pi) = \sum_{i=1}^{N} \sum_{s} P_i(s|\pi) \mathbb{E}_{a \sim \pi(s,i)} R(s, a)$ where $P_i(s|\pi)$ is the probability of arriving in $s$ at time step $i$ with policy $\pi$. RL aims to learn a policy $\pi_\theta$, parameterized by $\theta$, to maximize the reward.

Policy gradient is the leading class of algorithms to train policy networks. Intuitively, policy gradient optimizes the policy network by following the gradient of the expected reward regarding the policy parameters. The policy gradient is typically estimated on a finite collection of paths (i.e., trajectories) which are the past history of interactions between the policy and the environment. There are a lot of ways to estimate the policy gradient, including REINFORCE (Sutton et al., 1999), PPO (Schulman et al., 2017), DDPG (Lillicrap et al., 2015) and so on. The choice of policy gradients is independent of our method.

Generally speaking, there are two steps to estimate the policy gradient: (1) collecting a dataset of paths $D = \{\text{Path}^j\}_{j=1}^{n}$, $\text{Path}^j = \{(s_i^j, a_i^j, r_i^j := R(s_i^j, a_i^j)), i = 1, ..., N\}$ denotes the $j$-th path. (2) estimating policy gradient on $D$. Usually, $D$ should be sampled from the latest policy or be the collection of all past histories. For convenience, let $D_i$ denote the data at time step $i$ in $D$. As we will discuss in Sec. 3, this way of constructing $D$ turned out to be less sample efficient for high-dimensional SDE policy.

As for gradient estimation, we consider two popular algorithms: REINFORCE, which allows us to get unbiased estimation from samples, and DDPG, which directly calculates policy gradient through back-propagation from critic. Many popular algorithms are developed upon REINFORCE and DDPG. REINFORCE and DDPG calculate policy gradients as follows:

- **REINFORCE**: REINFORCE updates model parameters as:

$$\theta \leftarrow \theta + \eta \mathbb{E}_{j \sim U([n])} \mathbb{E}_{i \sim U([N])} \left( \nabla_\theta \log \pi_\theta(a_i^j | s_i^j, i) \sum_{i'=i}^{N} r_{i'}^j \right), \tag{2}$$

where $U(\cdot)$ denote the uniform distribution. In practice, a critic network $Q_\phi(s_i^j, a_i^j, i)$ can be used to approximate $\sum_{i'=i}^{N} r_{i'}^j$.

- **Deep deterministic policy gradient (DDPG)**: DDPG trains the actor $\pi_\theta$ as

$$\theta \leftarrow \theta + \eta \mathbb{E}_{j \sim U([n])} \mathbb{E}_{i \sim U([N])} \nabla_\theta Q_\phi(s_i^j, a, i), a \sim \pi_\theta(\cdot | s_i^j, i) \tag{3}$$

where the gradient from $a$ to $\theta$ is typically calculated by the reparameterization trick (Schulman et al., 2015).

Let $y_i^j = \sum_{i'=i}^{N} r_{i'}^j$, we train $Q_\phi$ by minimizing the loss:

$$\mathcal{L}(\phi) = \sum_{j=1}^{n} \sum_{i=1}^{N} (y_i^j - Q_\phi(s_i^j, a_i^j, i))^2 \tag{4}$$

The critic may also be trained by Bellman difference loss (Bellman, 1966).

## 2.2 Generative modeling via Stochastic differential equations

Generative modeling aims to approximate an unknown distribution $p_0$ given samples. Neural networks parameterized by SDEs turned out to be a super powerful tool for generative modeling. The leading generative SDEs are known as diffusion models. The core idea behind diffusion models is that we can construct a forward process by injecting noise into samples from $p_0$ and directly learn the corresponding backward generative SDE by maximum likelihood methods like score matching. More specifically, for any distribution $p_0$, we can construct the forward process:

$$dx = f(x,t)dt + g(t)d\omega, \tag{5}$$

which induces the marginal distribution $p_t(x)$ at time $t$. According to (Anderson, 1982), there is a corresponding backward process.

$$dx = (f(x,t) - g^2(t)\nabla_x \log p_t(x))dt + g(t)d\bar{\omega}. \tag{6}$$

Eq. 5 and Eq. 6 share the same marginal distribution and $\log_x p_t(x)$ is known as the score function. And we can learn $\epsilon_\theta(x_t, t)$ to approximate $\nabla_x \log p_t(x_t)$ by minimizing the score-matching loss:

$$\mathcal{L}_{score}(\theta) = \mathbb{E}_{t \sim U(0,1)} \mathbb{E}_{x_0} \mathbb{E}_{x_t \sim q_{t0}(x_t|x_0)} \| \epsilon_\theta(x_t, t) - \nabla_{x_t} \log p_{t0}(x_t|x_0) \|^2, \tag{7}$$

where $q_t$ denotes the marginal distribution of estimated backward SDE induced by the model $\epsilon_\theta$ in Eq. 1 at time $t \in [0, 1]$. And $p_{tt'}(x_t|x_{t'})$ (resp. $q_{tt'}(x_t|x_{t'})$) denotes the conditional distribution of $x_t$ given $x_{t'}$ in forward (resp. estimated backward) SDE. In this case, $\pi_\theta(x_t, t) := f(x_t, t) - g^2(t)\epsilon_\theta(x_t, t)$.

Once $\epsilon_\theta$ is fixed, generating samples can be done by using solvers (Song et al., 2021a; Lu et al., 2022). The idea of learning backward process from a given forward process is further to ODEs using flow matching (Lipman et al., 2023). And the forward process will play a central role in our method.

## 2.3 SDE as Markov Decision Process

We can consider the SDE as an MDP with infinitely small time steps which is extended from Black et al. (2023). More specifically, in the limit $N \to \infty$, the discrete-time MDP from time 0 to $N$ becomes a continuous MDP from time 1 to 0 [1]. We map the SDE in Eq. 1 to a continuous MDP as:

---

[1] We let the time flow from 1 to 0 to make the notation consistent with that in diffusion models.

$$s_t \triangleq x_t, \quad \pi_\theta(s_t, t) \triangleq f(x, t) - g^2(t)\epsilon_\theta(x_t, t), \quad \rho_0 \triangleq \mathcal{N}(0, I),$$

$$s_{t-dt} \triangleq \pi_\theta(x_t, t)dt + g(t)d\bar{\omega}, \quad R(s_t, a_t, t) \triangleq \begin{cases} 0, & \text{if } t > 0, \\ R(s_0), & \text{if } t = 0, \end{cases} \quad (8)$$

Therefore, we may directly apply existing policy gradients to SDEs. However, as we will analyze in the next section, the naive application of policy gradients will result in an unstable training process and unsatisfactory sample complexity.

## 3 CHALLENGE OF APPLYING POLICY GRADIENT TO TRAIN SDEs

In this section, we take a close look at the practical issues of directly applying policy gradient to train SDEs. These issues are caused by the fact that we estimate the policy gradient on a finite set of trajectories $D$, which result in ill-defined and instable policy gradients. These challenges motivate us to regularize the SDE around a perturbation forward process and exploit the perturbation nature of the forward process to stabilize the training process. For the sake of simplicity, we focus on DDPG in this section, and the analysis for the REINFORCE algorithm is similar.

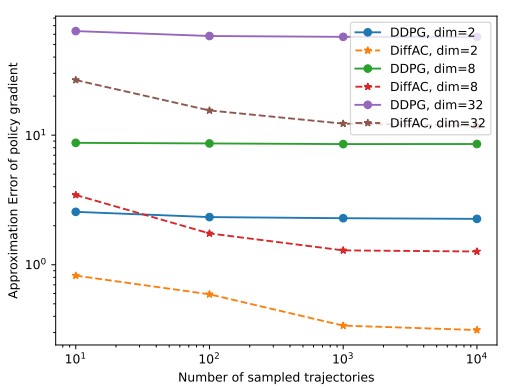

### 3.1 ILL-DEFINED POLICY GRADIENT

According to chain rule, the policy gradient in Eq. 3 is $\nabla_{\pi_\theta(x_t,t)} Q_\phi(x_t, \pi_\theta(x_t, t), t)\nabla_\theta \pi_\theta(x_t, t)$. Recall that $Q_\phi$ is trained by minimizing loss in Eq. 4. Since we do not have any direct training signal on $\nabla_{\pi_\theta(x_t,t)} Q_\phi(x_t, \pi_\theta(x_t, t), t)$, the gradient should be inferred from data around $\pi_\theta(x_t, t)$. Therefore, when there is insufficient data in the vicinity of $(x_t, \pi_\theta(x_t, t), t)$, it is difficult to reliably estimate the gradient $\nabla_{\pi_\theta(x_t,t)} Q_\phi(x_t, \pi_\theta(x_t, t), t)$. Hence, to obtain an accurate estimation for $\nabla_{\pi_\theta(x_t,t)} Q_\phi(x_t, \pi_\theta(x_t, t), t)$, it is essential to gather more samples around the data. As the

Figure 1: Comparison on the prediction error of policy gradients with respect to the number of trajectories under different settings of dimensionality. We evaluate $\mathbb{E}_{x_t} \|\nabla_{x_t} Q_\phi(x_t, \pi_\theta(x_t, t), t) - \nabla_{x_t} Q_{\phi^*}(x_t, \pi_\theta(x_t, t), t)\|$ where $\phi^*$ is trained on a large number of trajectories and $\phi$ is trained on a small number of trajectories. We can see the prediction error on policy gradient of our method is much lower than that of DDPG. Please refer to appendix for more details of this experiment.

dimensionality increases, the demand for data becomes greater, leading to unsatisfactory sample complexity.

To see this, we present a toy example to describe the relationship between the estimation error on policy gradient and the number of sampled trajectories. We compare the conventional DDPG against our method described in the next section. We can see that given the same number of sampled trajectories, our method provides a better policy gradient estimation.

### 3.2 UNCONTROLLED BEHAVIOR FOR LOW DATA DENSITY REGION

If $(x_t, t)$ is distant from the data distribution, the behavior of $\pi_\theta(x_t, t)$ may be uncontrolled, as the loss in Eq. 3 does not define the behavior of $\pi_\theta(x_t, t)$ in regions with low data density. Training $\pi_\theta$ solely on the limited collected paths inevitably encounter sparse data areas, and the SDE encounters these data-scarce regions. In such cases, the policy selects actions indiscriminately, consequently diminishing the effectiveness of obtaining feedback signals from the environment.

To give a better illustration, let's consider a one-step MDP in $\mathbb{R}^2$, in which the agent observes one state $s$, outputs an action $a$ and the MDP terminates. We suppose the reward is $-\|s+a\|^2$. It is easy to see that the optimal policy should always output $-s$. Consider the case that we only train $\pi_\theta(s,1)$ with $s$ such that each dimension is randomly sampled from interval $(1,2)$, and remains the policy for other regions untrained. We present the result in Fig. 2. We can see that the policy only works well on the data-intensive area and poorly behaves in data-scarce areas. While this example seems to be contrived, when the policy is an SDE, it is inevitable that the simulation runs into data-scarce regions. Therefore, the policy might have uncontrolled behavior and further hurt the sample complexity.

Figure 2: The reward of $\pi_\theta(x_1,1)$ in different region. The policy receives high reward in bright colored region. We can see that the policy only works well on region close to training set.

## 4 METHOD

To resolve the above challenge, we introduce a novel actor-critic policy gradient algorithm which is specifically designed for SDEs. The key is to constrain the SDE to be consistent with its associated forward perturbation process. Recall that the process of estimating the policy gradient consists of two stages: (1) generating a dataset $D \sim \pi_\theta$, and (2) estimating the policy gradient based on $D$. We will discuss these two parts respectively.

### 4.1 GENERATING SAMPLES

As presented in Sec. 3, the challenges stem from the fact that the policy gradient is estimated using a finite set $D \sim \pi_\theta$ of sampled paths. Accurate estimation of the policy gradient occurs only when $D$ is relatively large, leading to unsatisfactory sample complexity. Our key observation is that by ensuring the SDE aligns with its associated perturbation process, the policy gradient estimation can be made more robust.

**Definition 1** (Consistent SDE). *An SDE is the associated forward perturbation process for the backward SDE in Eq. 1 if and only if $q_0 = p_0$. Furthermore, if $q_t = p_t$ for all $t \in [0,1]$, then the forward SDE and backward SDE are called consistent.*

It is noteworthy that a backward SDE is consistent with its associated forward SDE if and only if the score loss in Eq. 7 is minimized at $\epsilon_\theta$. More importantly, when the backward SDE is consistent, we have a more robust and efficient way to sample from $q_t$ and further estimate the policy gradient more accurately.

---

**Algorithm 1** DiffAC-v1

**Input:** Initialized actor $\epsilon_\theta$ and critic $V_\phi$, reward function $R(\cdot)$, $D_0 = \emptyset$
**Output:** $\theta, \phi$
1: **for** each iteration **do**
2:     Sample $D_0 \leftarrow \{x_0^j\}_{j=1}^n$ from Eq. 1.
3:     Train $\epsilon_\theta$ by minimizing the score matching loss.
4:     Train $V_\phi$ according to Eq. 10.
5:     **for** each iteration **do**
6:         Calculate policy gradient $\mathcal{PG}(\theta)$ according to Eq. 11 or Eq. 12.
7:         $\theta \leftarrow \theta + \eta \mathcal{PG}(\theta)$.
8:     **end for**
9: **end for**

---

**Lemma 1.** *If the SDE defined by $\epsilon_\theta$ is consistent, let $x_t \sim p_{t0}(x_t|x_0)$ where $x_0 \sim q_0(x_0)$. Then, we have $x_t \sim q_t(x_t)$.*

*Proof.* The proof is straightforward. If the SDE is consistent, we have

$$\int_{x_0} q_0(x_0)p_{t0}(x_t|x_0)dx_0 = \int_{x_0} p_0(x_0)p_{t0}(x_t|x_0)dx_0 = \int_{x_0} p(x_t,x_0)dx_0 = p_t(x_t) = q_t(x_t).$$

$\square$

**Generating $\tilde{D}_t$ for policy gradient estimation by perturbing** $D_0$: Considering the initial dataset $D_0 = \{x_0^j \sim q_0(x_0)\}_{j=1,...,n}$, it is feasible to directly produce an arbitrarily large set $\tilde{D}_t = \{x_t^j\}_{j=1,...,N}$, where $x_t^j \sim p_{t0}(x_t|x_0)$ for a given $x_0 \in D_0$. Lem. 1 demonstrates that $x_t^j \sim q_t(x_t)$, thereby implying that it is possible to estimate the policy gradient on samples perturbed from $D_0$.

Intuitively, the construction of $\tilde{D}_t$ mitigates the practical issues in Sec. 3 as follows: Firstly, it is evident that the perturbation process encompasses the entire space, resulting in a well-defined policy gradient. Although the samples from $\tilde{D}_t$ are not entirely independent, Fig. 1 indicates that policy gradients estimated on $\tilde{D}_t$ exhibit accuracy when $n$ is comparatively small. Secondly, for consistency SDEs, the distribution of training data is guaranteed to have the same distribution. Therefore, the probability for a consistent SDE to run into data-scarce region is relatively small.

## 4.2 ESTIMATION OF POLICY GRADIENT

Given the process of generating samples mentioned above, it is easy to extend the policy gradient in Eq. 2 and 3. We consider the actor-critic framework, where a critic is trained to predict the cumulative reward and the actor is trained based on the reward signal provided by the critic. We observe that with SDE policy, it is more convenient to train the critic $V_\phi(x_t, t)$ to predict the reward given $x_t$ rather than $Q_\phi(x_t, a_t, t)$. Hence, we will focus on the training of $V_\phi$ instead. The combination of the actor and critic training procedures, as well as the score-matching loss and the construction of $\tilde{D}_t$ in Sec. 4.1, results in our first actor-critic algorithm presented in Alg. 1. Since the forward perturbation process draws inspiration from diffusion models, we name our algorithm Diffusion Actor-Critic (DiffAC).

**Critic Training**: For any consistent SDE $\epsilon_\theta$, $V_{\phi^*}(x_t, t) = \mathbb{E}_{x_0 \sim q_{0t}(x_0|x_t)} R(x_0)$ if and only if

$$\phi^* = \arg\min_\phi \mathbb{E}_t \mathbb{E}_{x_0 \sim q_0(x_0)} \mathbb{E}_{x_t \sim p_{t0}(x_t|x_0)} (V_\phi(x_t, t) - R(x_0))^2. \tag{9}$$

The derivation of Eq. 9 is straightforward. Therefore, to train the critic, we just need to perturb $D_0$ to get $\tilde{D}_t$ and minimize the loss in Eq. 9, leading to:

$$\text{(Critic loss):} \qquad \frac{1}{n} \sum_{j=1}^n (V_\phi(x_t^j, t) - R(x_0^j))^2, \tag{10}$$

where $x_t^j \sim p_{t0}(\cdot|x_0^j)$, $x_0^j \in D_0$, and $R(x_0^j)$ is the reward of $x_0^j$.

**Actor Training**: We now proceed to extend the REINFORCE and DDPG algorithms to SDEs. Consider the sample $\bar{x}_t^j$, which is generated by simulating Eq. 1 for an infinitesimal time step starting from $x_t^j$, and can be represented as $\bar{x}_t^j \sim \pi_\theta(x_t^j, t)dt + \sigma_t d\bar{\omega}$. Furthermore, let $\mathcal{P}_\theta(\bar{x}_t|x_t)$ denote the density of this sample. We have:

**Theorem 1.** *For SDE policy, we can estimate the policy gradient as:*

$$\text{(SDE-REINFORCE)}: \qquad \mathcal{PG}(\theta) \leftarrow \frac{1}{n} \sum_{j=1}^n \nabla_\theta \log \mathcal{P}_\theta(\bar{x}_t^j|x_t^j) V_\phi(\bar{x}_t^j, t - dt), \tag{11}$$

$$\text{(SDE-DDPG)}: \qquad \mathcal{PG}(\theta) \leftarrow \frac{1}{n} \sum_{j=1}^n \nabla_\theta V_\phi(\bar{x}_t^j, t - dt), \quad \bar{x}_t^j \sim \mathcal{P}_\theta(\bar{x}_t^j|x_t^j), \tag{12}$$

*where $x_t^j \sim p_{t0}(x_t^j|x_0^j)$ and $x_0^j$ is sampled from $D_0$.*

In practice, the infinitesimal time step can be replaced by discrete time steps in solvers. We suppose that an SDE solver generates simulates Eq. 1 from a iterative procedure: $x_{\tau+1} = \alpha_\tau(x_\tau - \beta_\tau \pi_\theta(x_\tau, t_\tau)) + \zeta_\tau z_\tau$ where $z_\tau \sim \mathcal{N}(0, I)$, $\tau = 1, \ldots, T$, $t_{\tau+1} < t_\tau$ with $t_1 = 1, t_T = 0$, $x_1 \sim \mathcal{N}(0, I)$, $\alpha_\tau, \beta_\tau, \zeta_\tau$ are specified by solvers. Many popular solvers for diffusion models fall into this formulation, e.g., DDIM (Song et al., 2021a), DDPM (Ho et al., 2020). For discretization of time for SDE policy, we can similarly replace $\bar{x}_t^j$ or $t - dt$ in Eq. 12 and 11 with $x_\tau$ or $t_\tau$ respectively.

### 4.3 A PRACTICAL IMPLEMENTATION

In Alg. 1, we first run score-matching to make sure $\epsilon_\theta$ is consistent, and then apply policy gradient to optimize the reward. However, during the training step, the model may rapidly forget the knowledge learned during score-matching , which leads to frustrating inconsistency. Therefore, we introduce an additional policy which is trained to maximize reward under the regularization of score-matching policy to alleviate this inconsistency. The regularization is:

$$KL(\epsilon_{\theta'}, \epsilon_\theta, D) = \frac{1}{|D|} \sum_{x_t \in D} KL(\pi_{\theta'}(x_t, t), \pi_\theta(x_t, t)). \quad (13)$$

And we update our policy $\epsilon_{\theta'}$ as

$$\theta' \leftarrow \theta' + \eta_1 \mathcal{PG}(\theta') - \eta_2 \nabla_{\theta'} KL(\epsilon_{\theta'}, \epsilon_\theta, D), \quad (14)$$

where $\epsilon_\theta$ is trained on $D_0$ via score matching. Eq. 14 leads to Alg. 2. Moreover, we show that the objective in Alg. 2 also leads to the optimal policy.

**Lemma 2.** *Let $x_0^*$ denote the optimal point, that is, for any $x_0'$, $R(x_0') \geq R(x_0^*)$. Let $\epsilon_\theta^*$ denote the consistent SDE with $q_0 = \delta(x_0^*)$. Then, $\epsilon_\theta^*$ is the minimizer of loss in Eq. 14.*

*Proof.* The proof is straight-forward as $\epsilon_\theta^*$ is the minimizer for both terms in Eq. 14. ☐

## 5 RELATED WORK

**Diffusion Models and forward / backward SDEs** Diffusion models (Song & Ermon, 2019; Song et al., 2021b; Ho et al., 2020; Yang et al., 2022) have emerged as powerful tools in the field of generative models. Their primary objective is to maximize the likelihood of data distribution. To achieve this, Song et al. (2021b); Ho et al. (2020) construct forward stochastic differential equations (SDEs) by injecting noise and utilize their corresponding backward SDEs for generating samples. These backward SDEs can be efficiently trained using denoising score-matching (Vincent, 2011). SDEs exhibit a remarkable capability for modeling complex distributions and generating intricate images and structures. This potential of SDEs inspires us to explore their use as the foundation for policy networks.

---

**Algorithm 2** DiffAC-v2

**Input:** Initialized $\epsilon_\theta, \epsilon_{\theta'}$ and critic $V_\phi$, reward function $R(\cdot)$, $D_0 = \emptyset$

**Output:** $\theta'$

1: **for** each iteration **do**
2:     Sample $\{x_0^j\}_{j=1}^n$ by simulating $dx_t = \pi_{\theta'}(x_t, t)dt + \sigma_t d\bar{\omega}$.
3:     Sample $D_0 \leftarrow \{x_0^j\}_{j=1}^n$ from Eq. 1.
4:     Train $V_\phi$ according to Eq. 10.
5:     Train $\epsilon_\theta$ by score-matching in Eq. 7.
6:     **for** each iteration **do**
7:         Update $\theta'$ according to Eq. 14.
8:     **end for**
9: **end for**

---

**RL with Diffusion Models** Recently progress in RL has identified diffusion models as a powerful tool in policy modeling, due to its generative capability and standardized training process. In offline RL where we need to learn a policy from a given dataset, researchers typically exploit diffusion models to deal with heterogeneous datasets, generating in-distribution strategies or modeling complex strategy distribution (Janner et al., 2022; Wang et al., 2023; Ajay et al., 2023; Hansen-Estruch et al., 2023; Lu et al., 2023). In online RL, where we need to interact with an environment, Chen et al. (2023) uses diffusion models to model multi-modal distribution for exploration. Black et al. (2023); Fan et al. (2023) proposed to finetune a pretrained diffusion model with REINFORCE, but they didn't address the stability issue.

**Structure-based Drug Design** Structure-based drug design (SBDD) (Anderson, 2003) aims to generate ligand molecules given a protein binding site (i.e., protein pocket), which is a key tool in drug discovery. The ligand molecules are usually expected to have desired properties, such as high binding affinity to the target protein. Luo et al. (2021); Liu et al. (2022); Peng et al. (2022)

proposed to generate atoms (and bonds) of 3D ligands based on 3D protein pockets in an auto-regressive way. More recently, Guan et al. (2023); Lin et al. (2022); Schneuing et al. (2022) employed SE(3)-equivariant diffusion models for SBDD. To design molecules with desired properties (i.e., inverse design), Bao et al. (2023) proposed equivariant energy-guided stochastic differential equations (EEGSDE). We test our method on SBDD and achieve superior performance than EEGSDE and its stronger variants.

## 6 EXPERIMENTS

We demonstrate the effectiveness of our methods on structure-based drug design (SBDD) (Anderson, 2003). Here we apply our methods to promote the binding affinity of ligand molecules generated by diffusion models.

### 6.1 EXPERIMENTAL SETUP

**Dataset** Following the previous work Luo et al. (2021); Peng et al. (2022); Guan et al. (2023), we use the CrossDocked2020 dataset (Francoeur et al., 2020) for both training and optimization. We follow the same dataset pre-processing and splitting procedure as Luo et al. (2021). $100,000$ pocket-ligand pairs are used for training, and $100$ pockets are used for testing. The goal is to generate ligands that bind to the pockets in the test set with high binding affinity.

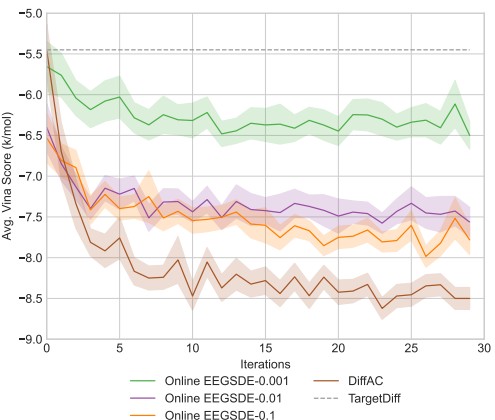

Figure 3: Optimization curves which show how average Vina Score of generated ligand molecules changes with the number of iterations.

**Evaluated Methods** We implement and evaluate some baselines and DiffAC: (1) AR (Luo et al., 2021). (2) Pocket2Mol (Peng et al., 2022). (3) TargetDiff (Guan et al., 2023). (4) EEGSDE (Bao et al., 2023). We implement the equivariant energy guidance on TargetDiff. The energy function is the same as the pre-trained critic that we have mentioned above. $\{0.001, 0.01, 0.1, 1\}$ are used as coefficients of energy guidance during sampling. (5) Online EEGSDE. To better leverage interactions with the environment, we online train the energy function as we do for DiffAC. Here we also follow the setting about the number of optimization iterations and the number of generated samples and updates of the critic in each iteration. Similarly, $\{0.001, 0.01, 0.1, 1\}$ are also used was guidance coefficients. (5) We implement DiffAC based on TargetDiff. The hyperparameters in our method are the same for all target pockets. Refer to Appendix B for more implementation details. We do not report the results of DDPO (Black et al., 2023) and DPOK (Fan et al., 2023) as they do not have satisfactory performances in our evaluation.

**Evaluation** Designing molecules with desired properties (i.e., molecular inverse design) is a fundamental and valuable task. Here we choose binding affinity as our target due to its importance in structure-based drug design. We employ AutoDock Vina (Eberhardt et al., 2021) to estimate the binding affinity of pairs of the protein pockets and the generated ligands, following the same setup as Luo et al. (2021); Guan et al. (2023). Optimizing Vina Score is a challenging task because not only the molecules themselves but also their chemical and spatial interaction with the 3D protein pockets need to be considered. We generate $100$ ligand molecules across $100$ pockets in the test set using TargetDiff and EEGSDE. For online EEGSDE and DiffAC, we first finish the online fine-tuning process on each pocket separately and use the best and the last checkpoint to generated $100$ ligand molecules, respectively. For all methods, we collect all generated molecules across $100$ test proteins and report the mean and median of Vina Score.

### 6.2 MAIN RESULTS

We evaluate all baselines and our method under the setting introduced in Sec. 6.1. As Tab. 1 shows, our method performs better than all other baselines. EEGSDE with proper energy guidance coefficient can indeed improve the property of generated molecules. And the online variants of EEGSDE can

Table 1: Summary of Vina Score of ligand molecules generated by all baselines and our method. Note that a smaller score is better. All online algorithms are tested using the best checkpoint (denoted as Best Run) and the last checkpoint (denoted as Last Run), respectively. The improvements (resp. deteriorations) compared with TargetDiff are higlighed in green (resp. red).The standard deviation is highlighted in blue.

| Method | Best Run | | Last Run | |
|---|---|---|---|---|
| | Avg. | Med. | Avg. | Meg. |
| Reference | -6.36 | -6.46 | - | - |
| AR | -5.75 ±1.39 | -5.64 | - | - |
| Pocket2Mol | -5.14 ±1.60 | -4.70 | - | - |
| TargetDiff | -5.45 ±2.46 | -6.30 | - | - |
| EEGSDE-0.001 | -5.66 ±2.78 (-0.20) | -6.51 (-0.21) | - | - |
| EEGSDE-0.01 | -6.40 ±2.61 (-0.95) | -7.05 (-0.75) | - | - |
| EEGSDE-0.1 | -6.53 ±3.08 (-1.08) | -7.35 (-1.05) | - | - |
| EEGSDE-1 | -3.30 ±1.59 (+2.15) | -4.67 (+1.63) | - | - |
| Online EEGSDE-0.001 | -7.17 ±1.86 (-1.72) | -7.16 (-0.86) | -6.50 ±2.47(-1.05) | -6.61 (-0.31) |
| Online EEGSDE-0.01 | -8.22 ±1.89 (-2.77) | -8.06 (-1.76) | -7.56 ±2.46 (-2.11) | -7.51 (-1.21) |
| Online EEGSDE-0.1 | -8.58 ±1.70 (-3.13) | -8.52 (-2.22) | -7.78 ±2.33 (-2.33) | -7.78 (-1.48) |
| Online EEGSDE-1 | -7.13 ±1.08 (-1.68) | -7.28 (-0.98) | -2.13 ±2.10 (+3.32) | -4.29 (+2.01) |
| **DiffAC** | **-9.07 ±1.99 (-3.62)** | **-9.04 (-2.74)** | **-8.50 ±2.11 (-3.05)** | **-8.38 (-2.08)** |

TargetDiff

| Vina Score: +2.00 | Vina Score: +0.42 | Vina Score: +4.33 | Vina Score: +2.36 | Vina Score: +1.24 | Vina Score: +1.11 |

DiffAC

| Vina Score: -12.70 | Vina Score: -10.42 | Vina Score: -13.04 | Vina Score: -9.72 | Vina Score: -7.47 | Vina Score: -9.26 |
| 2Z3H | 4G3D | 5Q0K | 5B08 | 4Q8B | 1D7J |

Figure 4: Examples of generated ligands. Carbon atoms in ligand molecules by TargetDiff (Guan et al., 2023) and DiffAC are visualized in green and cyan, respectively. Here we select some cases where TargetDiff easily generates unrealistic ligand molecules that clash with protein surfaces physically which usually leads to extremely bad Vina scores. DiffAC can sample realistic ligand molecules with high quality in these hard cases.

further improve the performance, which shows the benefits of online training the critic. Our method can even outperform online EEGSDE with the best energy guidance coefient by a large margin and achieve the best Avg. Vina Score over all methods for structure-based drug design, which demonstrates the effectiveness of DiffAC. We visualize examples of ligand molecules generated by TargetDiff and DiffAC on some hard cases in Fig. 4. As Fig. 3 shows, DiffAC converges faster than all other online optimization algorithms, which demonstrates the superiority of our method in terms of sample complexity and training efficiency. We also provide the optimization curves of each protein pocket in Appendix C. The experiments has revealed the great potential of our method in important real-world applications, such as drug discovery.

## 7 CONCLUSIONS

This paper proposes DiffAC, a stabilized policy gradient method for SDEs, and demonstrate its superiority on structure-based drug design. This is a general framework with great potential. In terms of future work, it would be interesting to apply this method to many valuable applications where user

preferences or design requirements can be specified, such as text-to-image generation, protein design, chip design, etc.

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

## A    TOY EXAMPLE IN SEC 3.1

We compare the error on $\nabla_{\pi_\theta(x_t,t)}Q_\phi(x_t,\pi_\theta(x_t,t),t)$. Let $\phi'$ denote the critic trained by Eq. 4 on $n$ trajectories and $\phi''$ denote the trained by Eq. 9 on $D_0$ with $|D_0| = n$. We evaluate $|\nabla_{\pi_\theta(x_t,t)}Q_{\phi'}(x_t,\pi_\theta(x_t,t),t) - \nabla_{\pi_\theta(x_t,t)}Q_{\phi'^*}(x_t,\pi_\theta(x_t,t),t)|$ and $|\nabla_{\pi_\theta(x_t,t)}Q_{\phi''}(x_t,\pi_\theta(x_t,t),t) - \nabla_{\pi_\theta(x_t,t)}Q_{\phi''^*}(x_t,\pi_\theta(x_t,t),t)|$ where $\phi'^*$ and $\phi''^*$ are trained on $10^5$ trajectories.

The architecture of the critic network is 3 layered-MLP with hidden dimemsion 256. We train each network for $10^5$ iteration with batchsize 256. And A For reward function, we use Rastrigin function which is a toy function, with many local minimas, designed for testing zero order optimization algorithm.

## B    IMPLEMENTATION DETAILS

In this section, we will describe the implementation of experiments in detail.

Guan et al. (2023) employed an SE(3)-equivariant diffusion model, named TargetDiff, for structure-based drug design. Given a protein binding site, TargetDiff generates the atom coordinates in 3D Euclidean space and atom types by iteratively denoising from a prior distribution. After the reverse (generative) process of the diffusion model, the chemical bonds of the generated ligand molecules are defined as post-processing by OpenBabel (O'Boyle et al., 2011) according to the distances and types of atom pairs. We use TargetDiff as the actor and strictly follows the setting in Guan et al. (2023), such as noise schedules, model architecture, training objectives, etc.

We first pretrain TargetDiff on the training set. After that, we use the pretrained TargetDiff to first sample 100 ligand molecules for each pocket in the test set and evaluate their binding affinity by oracle. We pretrain the critic, which predicts the binding affinity based on the perturbed samples, on the $10,000$ generated pocket-ligand pair data. The model architecture of the critic is almost the same with TargetDiff. The only difference is that the critic has an aggregation layer at last to output a scalar based on global features. We finetune the pretrained TargetDiff (Guan et al., 2023) for 30 iterations for each pocket in the test set, respectively. In each iteration, we sample 34 ligand molecules induced by the diffusion model (i.e., the actor), evaluate the binding affinity by oracle, and then online update the diffusion model (i.e., the actor) and train the policy and critic following Alg. 2. We keep all sampled molecules in $D_0$ which falls into the class of off-policy policy gradient.

We use Adam (Kingma & Ba, 2014) with `init_learning_rate=0.001, betas=(0.95, 0.999), batch_size=8` and `clip_gradient_norm=8.0` to pretrain TargetDiff (i.e., the actor) and the critic. We use Adam with `init_learning_rate=0.0003` for online updating the actor and critic. As for regularization in Eq. 14, we set $\eta_2 = 0.05$ for atom types and $\eta_2 = 0.00025$ for atom positions.

## C    OPTIMIZATION CURVES OF $100$ PROTEIN POCKETS

We plot the optimization curves of DiffAC for the pocket protein in the test separately in Fig. 5, 6, and 7. Given a pocket protein, at each iteration, the average Vina Score of the sampled ligand molecules in this iteration is plotted as a point in the figure. The curves show how the binding affinity change with the number of optimization iterations. Generally, in most cases, DiffAC performs better than the baselines.

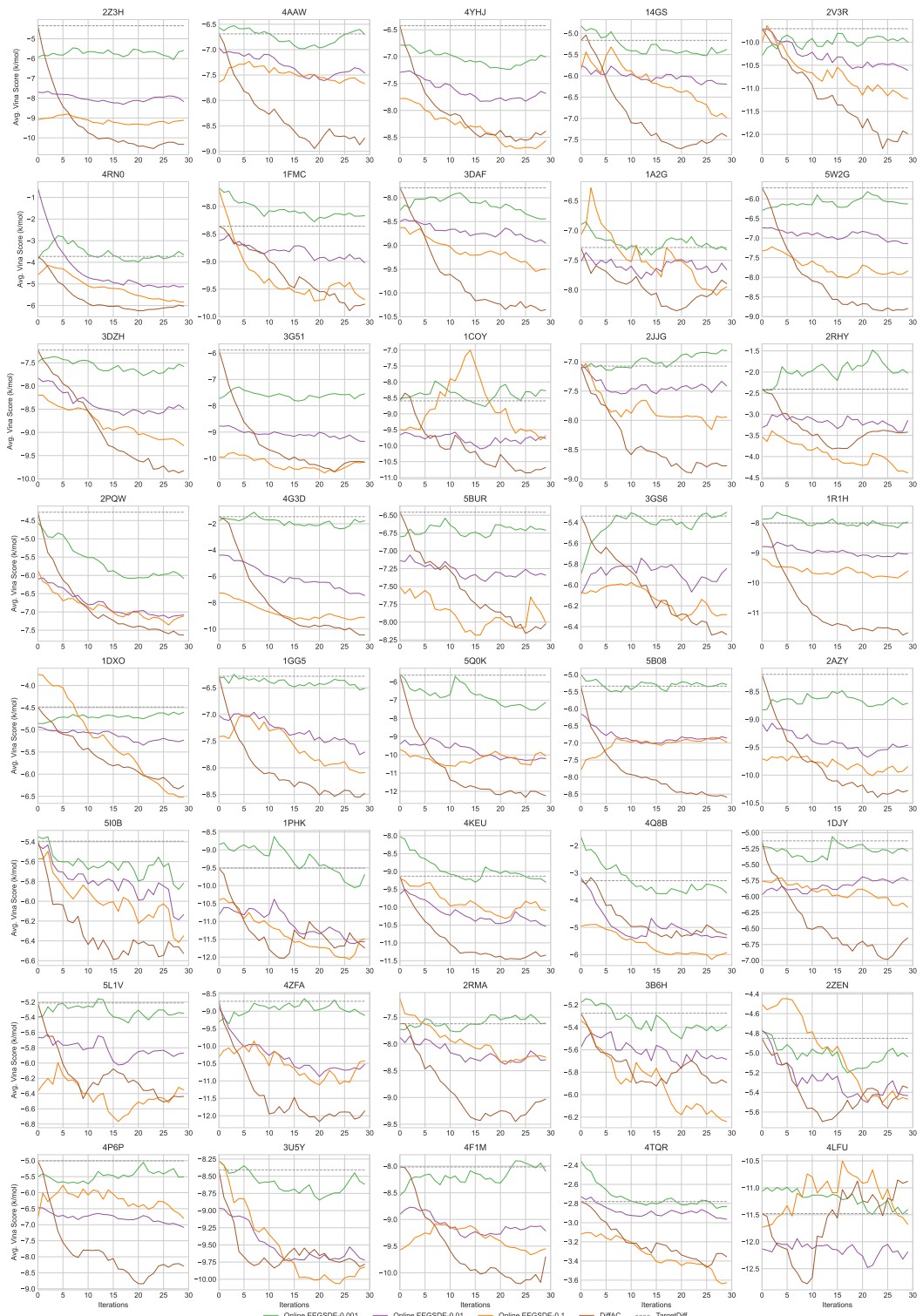

Figure 5: Optimization curves of the 1st to 40th protein pockets in the test set.

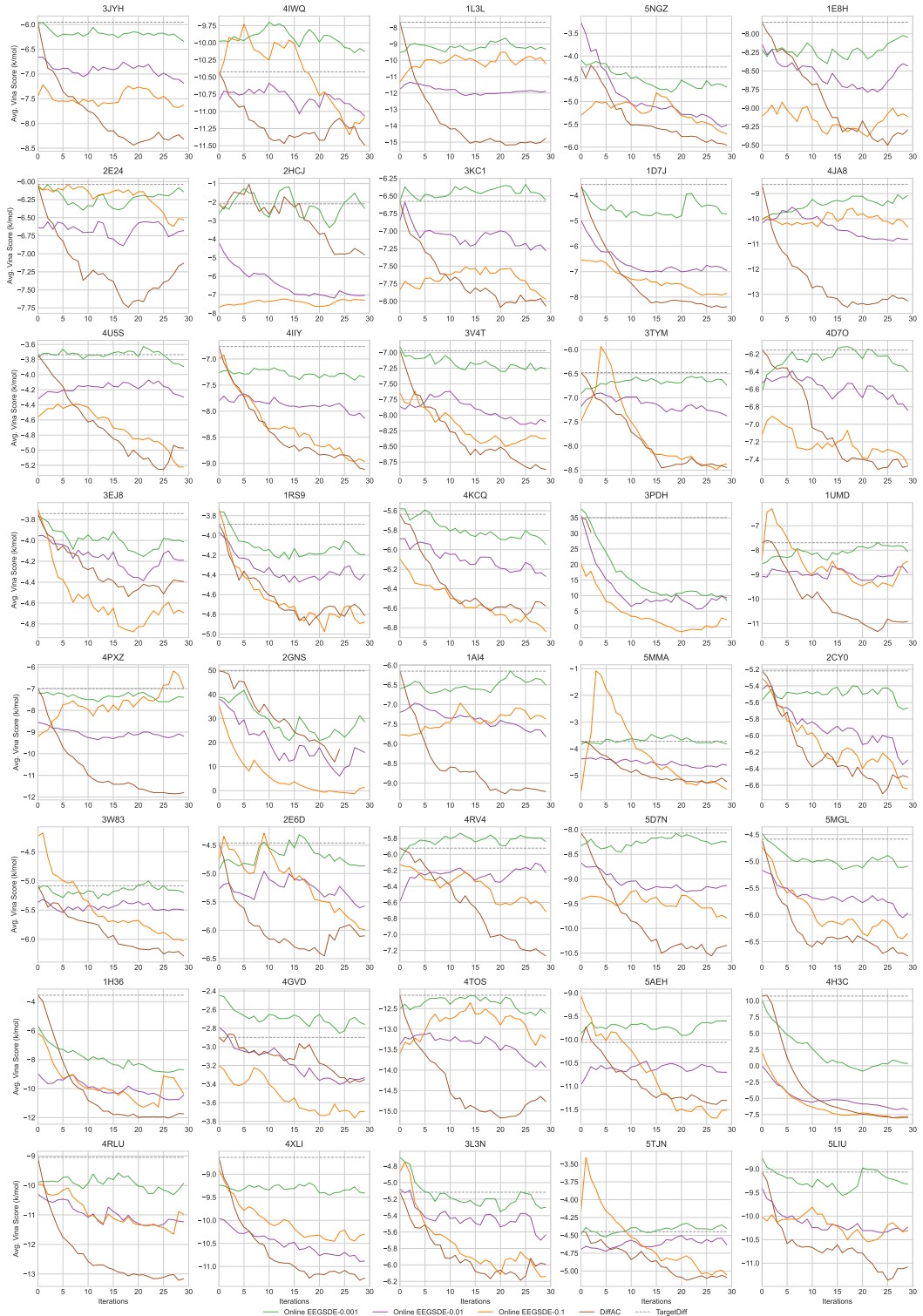

Figure 6: Optimization curves of the 41st to 80th protein pockets in the test set.

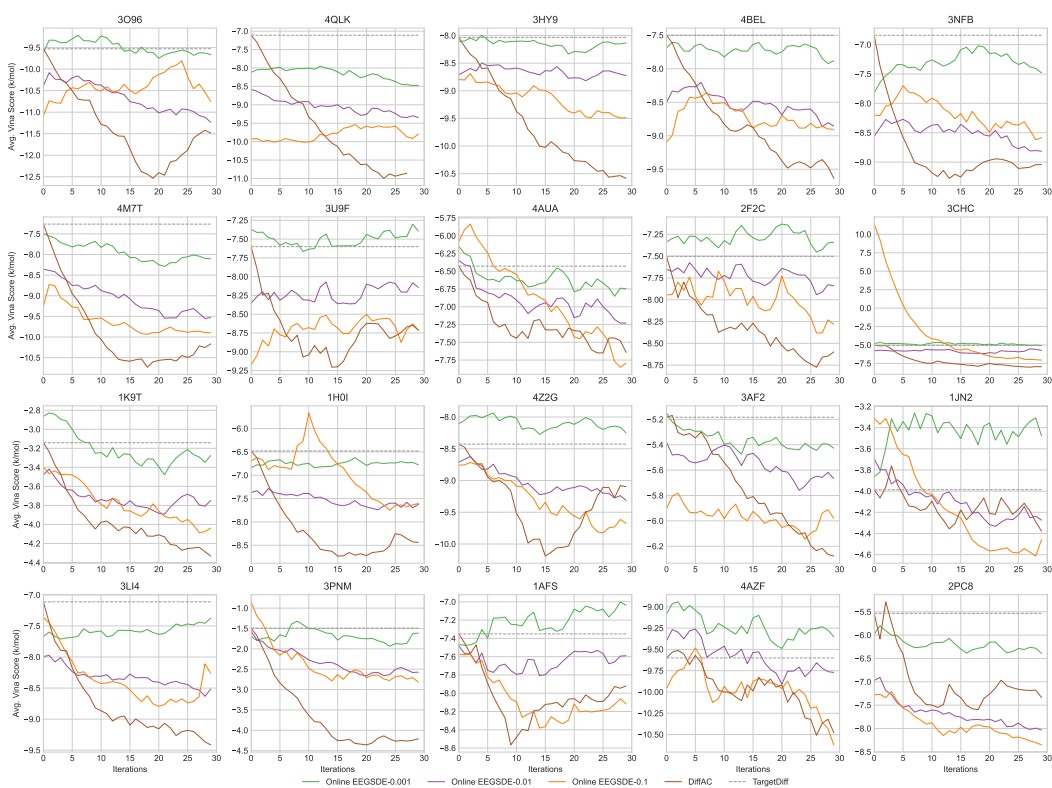

Figure 7: Optimization curves of the 81st to 100th protein pockets in the test set.

# D EXPERIMENTS ON TEXT-TO-IMAGE GENERATION

To demonstrate the generalizability of our method beyond the SBDD task, we also apply our method to text-to-image generation. In this experiment, we use DiffAC to fine-tune text-to-image generative models to better align with human preferences.

## D.1 EXPERIMENTAL SETUP

We use Stable Diffusion v1.5 (Rombach et al., 2022) as the baseline, which has been pre-trained on large image-text datasets (Schuhmann et al., 2021; 2022). For compute-efficient fine-tuning, we use Low-Rank Adaption (LoRA) (Hu et al., 2022), which freezes the parameters of the pre-trained model and introduces low-rank trainable weights. We apply LoRA to the UNet (Ronneberger et al., 2015) module and only update the added weights. For the reward model, we use ImageReward (Xu et al., 2023) which is trained on a large dataset comprised of human assessments of images. Compared to other scoring functions such as CLIP (Radford et al., 2021) or BLIP (Li et al., 2022), ImageReward has a better correlation with human judgments, making it the preferred choice for fine-tuning our baseline diffusion model. In practice, we use DiffAC (the REINFORCE version, i.e., Eq. 11) to fine-tune Stable Diffusion.

We also compare our method with DPOK (Fan et al., 2023). DPOK is a strong baseline that updates the pre-trained text-to-image diffusion models using policy gradient with KL regularization to maximize the reward. Notably, the difference between DPOK and our method is that DPOK estimates policy gradient with real trajectories sampled by backward process while our method estimates policy gradient with efficient forward process. And this difference is the key factor for stabler policy gradient.

We adopt a straightforward setup that uses one text prompt "A green colored rabbit" during fine-tuning and compares ImageReward scores of all methods. For both DPOK and our method, we perform 5 gradient steps per sampling step. The sampling batch size is 10 and the training batch size is 32.

## D.2 EXPERIMENTAL RESULTS

We plot the optimization curves of all methods as shown in Fig. 8. As the results indicates, our method can efficiently improve ImageReward scores and outperform baselines by a large margin.

We provide image examples as shown in Fig. 9. Stable Diffusion tends to generate images with obvious mistakes like generating a rabbit with a green background given the prompt "A green colored rabbit", while our method generates much more satisfying images that are well aligned with the given text prompt. The experiments on text-to-image generation along with structure-based drug design demonstrate the generalizability of our method and reveal its great potential on many real-world applications.

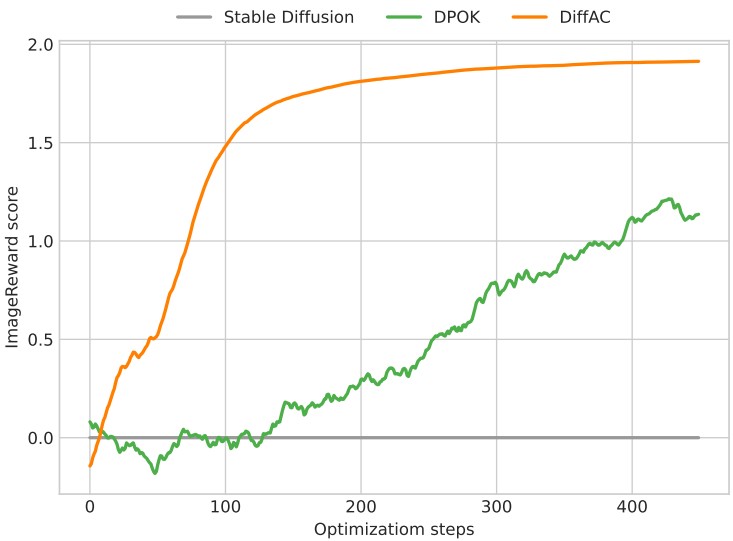

Figure 8: Optimization curves of ImageReward scores.

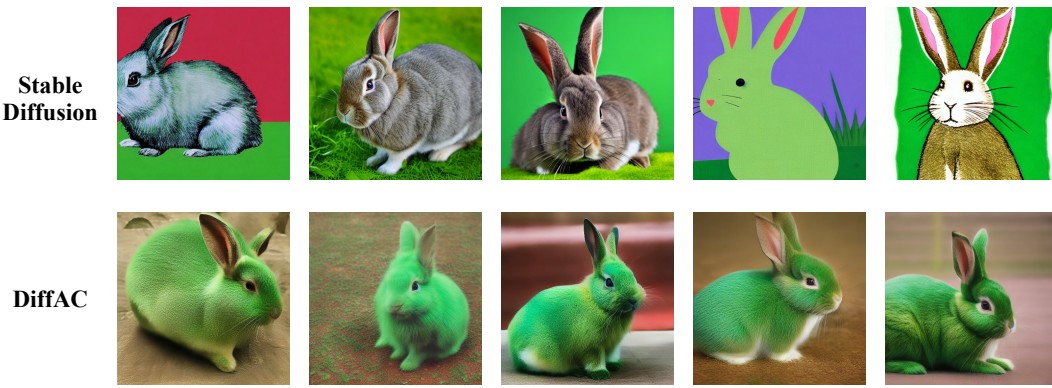

Figure 9: Example images generated by Stable Diffusion (Rombach et al., 2022) (top row) and our method (bottom row) givne text prompt "A green colored rabbit".