# OpenReview forum: "Stabilizing Policy Gradients for Stochastic Differential Equations by enforcing Consistency with Perturbation Process"
_ICLR.cc/2024/Conference — Submitted to ICLR 2024_

### Official Review · Reviewer_4LjP · 2023-10-28

**Soundness:** 3 good
**Presentation:** 3 good
**Contribution:** 3 good
**Rating:** 6
**Confidence:** 1

**Summary:**

This paper aims at mitigating the inherent instability associated with policy gradient estimation when employing Stochastic Differential Equation (SDE)-based policies rooted in the physical modeling of the diffusion model. This principle exhibits a broad applicability, adaptable to diverse policy gradient techniques. Empirical findings from experiments demonstrate a substantial outperformance of benchmark models, underscoring the considerable promise of the proposed approach.

**Strengths:**

This paper offers a simple principle to alleviate the instability of the policy gradient estimation with SDE-based policy from the physical modelling of the diffusion model itself. The principle is general to be instantiated to various policy gradient methods. The experiment results  significantly outperform benchmarks, which shows the convincing potential of the proposed method.

**Weaknesses:**

Although the proposed method appears to have the potential for instantiation in REINFORCE and DDPG, empirical evidence regarding the improvements from the perturbation process sampling remains unclear for both of these methods. While the authors have explained the necessity of the perturbation process using toy examples in Section 3, it would be more instructive and convincing if they could illustrate the quality of policy gradient estimation for both the vanilla policy gradient method and the perturbation process-enhanced versions.

Furthermore, the experimental results are limited to the SBDD task, which does not fully demonstrate the generalizability of the proposed method in a more convincing manner.

**Questions:**

Could the authors provide a more detailed explanation of the definition of the perturbation process and ensure consistency in Definition 1? I am not an expert in diffusion models, and at this point, I can only regard the perturbation process as a mechanism capable of generating realistic sample paths for the estimator. A more intuitive definition and explanation would greatly enhance understanding for readers like myself who may not be well-versed in this area.

---

> ### Author Response · Authors · 2023-11-21
> **Response to Reviewer 4LjP (1/2)**
>
> Thanks for your positive feedback. Please see below for our responses to the comments.
>
>
>
> **Q1: Additional experiments on text-to-image generation.**
>
> **"Furthermore, the experimental results are limited to the SBDD task, which does not fully demonstrate the generalizability of the proposed method in a more convincing manner."**
>
> A1: As the reviewer suggests, we demonstrate the generalizability by applying the proposed method to text-to-image generation task.
>
> We fine-tune the Stable Diffusion model [1] using ImageReward [2], an open-source reward model trained on a large dataset comprised of human assessments of (text, image) pairs. In our experiment, we use DiffAC as the fine-tuning algorithm to maximize the ImageReward score, which is the proxy for human preference.
> We also compare the our method with DPOK [3], a strong baseline that fine-tunes diffusion models by policy gradient with KL regularization. See the results in the table below.
>
> |    Methods              | ImageReward score ($\uparrow$) |
> |------------------|-------------------|
> | Stable Diffusion | 0.0               |
> | DPOK             | 1.3               |
> | DiffAC           | 1.9               |
>
> The experimental results show that our method efficiently achieves strong text-image alignment while maintaining high image fidelity. Please refer to Appendix D for more experimental details and results in our revised manuscript.
>
>
> **Q2: Performances of vanilla policy gradients.**
>
> **"Although the proposed method appears to have the potential for instantiation in REINFORCE and DDPG, empirical evidence regarding the improvements from the perturbation process sampling remains unclear for both of these methods."**
>
> A2: Thanks for your comments. Actually, before investigating DiffAC, we evaluated vanilla RL algorithms including REINFORCE and DDPG, and they turned out to be worse than EEGSDE, therefore we use EEGSDE as the baseline. And this motivated us to develop DiffAC. Moreover, in our new experiments on text-to-image generation, we directly compared DiffAC+REINFORCE against an improved version of REINFORCE [3] with KL regularization. And DiffAC significantly outperforms the baseline.
>
> **Q3: Quality of policy gradient estimation for both the vanilla policy gradient method and the perturbation process-enhanced version.**
>
> **"While the authors have explained the necessity of the perturbation process using toy examples in Section 3, it would be more instructive and convincing if they could illustrate the quality of policy gradient estimation for both the vanilla policy gradient method and the perturbation process-enhanced versions."**
>
> A3:
> We have actually illustrated the quality of policy gradient estimation for both methods in Section 3.1.
>
> As shown in Figure 1, we have demonstrated that the superiority of the perturbation-enhanced version in terms of quality of policy estimation by showing that its estimation error is lower in the toy example.
>
> More specifically, since the analytic policy gradient is not available, we view the policy gradient estimated with a sufficiently large number of samples as the ground truth. And we calculate the estimation error of policy gradient with different numbers of samples for both methods. We find that the perturbation-enhanced method consistently outperforms the vanilla one, especially when the number of samples is limited. The reason behind this is that the perturbation process effectively covers the whole action space and makes the actor fully utilize the limited samples. This toy experiment shows the motivation and main idea of our research.

---

> ### Author Response · Authors · 2023-11-21
> **Response to Reviewer 4LjP (2/2)**
>
> **Q4: More detailed explanation of the definition of perturbation process and consistency.**
>
> **"Could the authors provide a more detailed explanation of the definition of the perturbation process and ensure consistency in Definition 1? I am not an expert in diffusion models, and at this point, I can only regard the perturbation process as a mechanism capable of generating realistic sample paths for the estimator. A more intuitive definition and explanation would greatly enhance understanding for readers like myself who may not be well-versed in this area."**
>
> A: Thanks for your suggestions. We have formally discussed the SDE version of forward and backward process in Section 2.2. Now, we'd like to provide its discrete-time version of forward and backward process, which might be easier to understand for readers not familiar with diffusion models. The SDE version is essentially extended from DDPM by [4]. The following introduction follows DDPM [5].
>
> Let $q\_0(x\_0)$ denote an arbitrary distribution, we have
>
> 1. The forward (perturbation) process: $p(x\_{1:T}|x\_0) = \prod\_{t=1,...,T}p(x\_t|x\_{t-1})$ where $p(x\_t|x\_{t-1})=\mathcal{N}(x\_t|\sqrt{1-\beta\_t} x\_{t-1}, \beta\_t I)$ is a Gaussian distribution. $\beta\_t$ are pre-defined parameters which is linearly scheduled from $\beta\_1=1\times10^{-4}$ to $\beta\_T=0.02$ in DDPM. It is noteworthy that there is no trainable parameters in the forward process and for sufficiently large $T$, $p(x\_T| x\_0)\approx \mathcal{N}(0, I)$ the standard Gaussian distribution. Moreover, it is obvious that $p(x\_t | x\_0)$ is also a closed-form Gaussian process. In DiffAC, we exploited the forward process to stabilize the policy-gradient estimation.
>
> 2. The parameterized backward process: $q\_{\theta}(x\_{0:T})=q\_T(x\_T)\prod\_{t=T,...,1}q\_{\theta}(x\_{t-1}|x\_t)$ where $q\_T(x\_T)=\mathcal{N}(0, I)$ is the standard Gaussian distribution. And $q\_{\theta}(x\_{t-1}|x\_t)=\mathcal{N}(x\_{t-1}|\mu\_\theta(x\_t,t),\sigma\_\theta(x\_t, t))$. Generally speaking, the score matching loss is to minimize KL divergence between the marginal distributions of the forward and the backward process.
>
> 3. Consistency: once the backward process is fixed, we can define an associated forward process by letting  $p\_0(x\_0):=q\_0(x\_0)=\mathbb{E}\_{x\_T}q\_\theta(x\_0|x\_T)$. A backward process is called consistency if the associated forward process has the same marginal distribution with the backward process at every time step $t=0,...,T$. And the backward process is consistent if and only if the score matching loss is minimized.
>
> **References:**
>
> [1] Rombach, Robin, et al. High-resolution image synthesis with latent diffusion models. CVPR (2022).
>
> [2] Xu, Jiazheng, et al. Imagereward: Learning and evaluating human preferences for text-to-image generation. NeurIPS (2023).
>
>
> [3] Fan, Ying, et al. DPOK: Reinforcement Learning for Fine-tuning Text-to-Image Diffusion Models. NeurIPS (2023).
>
> [4] Song, Yang, et al. Score-based generative modeling through stochastic differential equations. ICLR (2021).
>
> [5] Ho, Jonathan, Ajay Jain, and Pieter Abbeel. Denoising diffusion probabilistic models. NeurIPS (2020).

---

> > ### Comment · Reviewer_4LjP · 2023-11-22
> >
> > The responses address the my concerns, and I would like to maintain positive scores for this work. However, as I am not an expert in this field, I remain the low confidence in my opinion.

---

### Official Review · Reviewer_QQeh · 2023-10-29

**Soundness:** 2 fair
**Presentation:** 2 fair
**Contribution:** 2 fair
**Rating:** 5
**Confidence:** 3

**Summary:**

Deep neural networks parameterized stochastic differential equations, which are especially important for their use in generative modeling, are trained by maximizing likelihood of training data. To use this for generating samples that maximize a user-desired criteria, reinforcement learning can be used for training. However, policy gradient-based optimization methods for reinforcement learning suffer when data is scarce, leading to instability and increasing sample complexity during training. The authors propose to mitigate this problem by making the trained SDE consistent with the associated perturbation process, which is global and covers wider space than training samples. The authors propose actor-critic policy gradient algorithm, and showcase their method’s efficacy on the problem of structure based drug design.

**Strengths:**

The authors described the problem and motivation in their abstract and introduction. The section about the challenges of applying policy gradient to SDEs (section 3) demonstrates that the author makes an effort to show motivating examples. The figure captions are detailed, and the related works are cited.

**Weaknesses:**

After reading the paper, I am not sure what the authors are trying to achieve. It seems to me that SDE is a tool for distribution modeling (modeling maximum likelihood), and they would like to use this approach to maximize some "rewards". If this is what the authors are trying to do, why should I care about distribution modeling? I mean, how does this improve standard optimization approaches, such as local search, monte-carlo tree search, reinforcement learning, etc? Why do I need to use this fancy SDE idea to search for molecules of some good properties? I think the authors have an intricate problem to solve, but the presentation is far from clear.

The authors need to make the equations and their associated explanations easy to read for general readers. The organization of the paper can be improved to increase clarity. Here are a few questions that the authors may consider clarifying in their revised manuscript. For example:

1.	In equation 1, what is sigma_t?
2.	Why is it natural to train SDEs with RL? What if the training data only consist of samples with desired characteristics? Do we still need RL in that case? The authors cited relevant work, but the basic idea/counterexample should be mentioned here (in a sentence)
3.	In equation 5, where is p_t as mentioned afterwards?
4.	Figure 1 and 2 needs to be further up in the paper, as they are the motivating examples for authors’ rationale of the paper.
5.	In section 2.3, it is better to state that the MDP is mapping to Equation 5 rather than Equation 1.
6.	In definition 1, what is p0,q0? There are no such symbols in Equation 1.

**Questions:**

See the weaknesses section.

---

> ### Author Response · Authors · 2023-11-21
> **Response to Reviewer QQeh (1/2)**
>
> Thanks for your constructive feedback. Please see below for our responses to the comments.
>
> **Q1: What is the goal? Why involve SDE in optimizing reward? Why integrate policy gradient with maximum likelihood?**
>
> **"After reading the paper, I am not sure what the authors are trying to achieve. It seems to me that SDE is a tool for distribution modeling (modeling maximum likelihood), and they would like to use this approach to maximize some "rewards". If this is what the authors are trying to do, why should I care about distribution modeling? I mean, how does this improve standard optimization approaches, such as local search, monte-carlo tree search, reinforcement learning, etc? Why do I need to use this fancy SDE idea to search for molecules of some good properties? I think the authors have an intricate problem to solve, but the presentation is far from clear."**
>
> A1: Thanks for your insightful comments. Our goal is to maximize the reward.
> Your concerns may come from our technique choices. Therefore, we'd like to address your concerns by elaborating the reasons for our choices.
>
> 1. the necessity of using SDE-based policy for molecular / image generation: when dealing with high-dimensional non-linear decision making problem, RL needs to exploit expressive parameterized policy to generate solutions with potentially high rewards. The parameterized policy can significantly prune the search space and accelerate the convergence rate, which has been validated on many applications, such as [1] and [2] for examples. When it comes to molecular optimization (as well as image optimization problem, please refer to our response to reviewer 4LjP's Q1 for our additional experiments on text-to-image generation), the SDE-based models are the SOTA of generating reasonable solution candidates. However, how to efficiently include SDEs as a policy into the standard RL workflow that you mentioned,  remains unclear. And this motivates us to explore this direction.
>
>
>
> 2. the motivation for training SDEs by integrating policy gradient with the score-matching (or MLE) loss: after deciding using SDE-based policy, we found that directly training SDEs with policy gradient is extremely unstable, please check section 3 for more details. Furthermore, we fortunately found that when the SDE is consistent, there is a novel approach to estimating the policy gradient by perturbation process, which is much more stable and more efficient. Thanks to the development of diffusion models, the score matching loss can ensure the SDE is consistent. Therefore, we choose to combine the score matching loss with the policy gradient approach.
>
>
> **Q2: Why RL if training data only consist of samples with desired properties?**
>
> **"Why is it natural to train SDEs with RL? What if the training data only consist of samples with desired characteristics? Do we still need RL in that case? The authors cited relevant work, but the basic idea/counterexample should be mentioned here (in a sentence)"**
>
>
> A2: Thanks for your comments. To address your concerns, let's take the molecular optimization problem as an example, where we want to find molecules with high binding affinity given a protein pocket.  Firstly, in many practical scenarios, the training set does not consist of data with desired characteristics. For the molecular optimization problem, TargetDiff is trained on protein-molecule pairs with Vina scores mainly ranging from -5 to -7. And with RL, we can significantly improve the average Vina score of generated molecules to -9.07. Secondly, we want to argue that even if the training data only consist of samples with desired properties, we still need RL.  Suppose we're testing TargetDiff on new proteins, the performance may be unsatisfactory (please check Figure 4), especially when the structure of the test data is different from those in the training set. In such cases, we need to exploit RL to further optimize the structure of the molecules.

---

> ### Author Response · Authors · 2023-11-21
> **Response to Reviewer QQeh (2/2)**
>
> **Q3: Background and notations of SDEs.**
>
> **"The authors need to make the equations and their associated explanations easy to read for general readers."**
>
> **"In equation 1, what is $\sigma\_t$?"**
>
> **"In equation 5, where is $p\_t$ as mentioned afterwards?"**
>
> **"In definition 1, what is $p\_0$, $q\_0$? There are no such symbols in Equation 1."**
>
> A3: We will carefully polish the manuscript according to your suggestions and add a more detailed discussion for RL, SDEs, and diffusion models in appendix in the next revision.
>  We here provide a more clear background and better notations of SDEs as follows:
>
>
> - (Equation 5) $dx = f(x,t) dt + g(t) dw$,
>  which is the forward SDE that corresponds to the perturbation process (i.e., the forward process of diffusion models).  $dw$ is a Wiener process.  $f(\cdot,t):\mathbb{R}^d\to\mathbb{R}^d$ is a vector-valued function called the drift coefficient of $x(t)$.  $g(t)$ is a scalar function of time and known as diffusion coefficient of the underlying dynamic.
>  - Following [3], applying Kolmogorov’s forward equation (Fokker-Planck equation) to an SDE, we can derive a probability ODE which describes how the marginal distribution ${p\_t}(x)$ evolves through time $t$.
> - (Equation 6) According to [4], an SDE as Equation 5 has a backward SDE: $dx= (f(x,t)- g^2(t)\nabla\_x\log p\_t(x))dt+g(t) d\bar{w}$, which shares the same marginal distribution $p_t(x)$ at time $t$. $d\bar{w}$ is the reverse Wiener process.
> - (Equation 1) $dx\_t=\pi\_\theta(x\_t,\theta)dt+g(t)d\bar{w}$, which is the approximated backward SDE parameterized by $\theta$. Conventionally, we use $\epsilon\_\theta(x\_t,t)$ to approximate the score $\nabla\_x\log p\_t(x)$. So we can let $\pi\_\theta(x\_t, \theta):=f(x\_t, t) - g^2(t)\epsilon\_\theta(x\_t, \theta)$. The approximated backward SDE corresponds to the generative process of diffusion models. Specifically, $x\_T$ is sampled from the prior distribution, evolves following this SDE, and arrives at $x\_0$. $x\_0$ is the generated sample.
> - $p\_0$ and $q\_0$ in Definition 1: You can find their definitions in section 2.2. Specifically, We denote the marginal distribution at time $t$ of Equation 5 and 1 as $p\_t$ and $q\_t$, respectively. Thus, $p\_0$ (resp. $q\_0$) is $p\_t$ (resp. $q\_t$) at time $t=0$.
>
>
> **Q4: View SDE as a MDP.**
>
> **"In section 2.3, it is better to state that the MDP is mapping to Equation 5 rather than Equation 1."**
>
> A4: Thanks for the suggestions. However, the forward process in Equation 5 is not an MDP as there is no action (please refer to Q4 for reviewer 4LjP for a more intuitive explanation of forward process). In Equation 1, we view $dx\_{t}=\pi\_\theta(x\_t,\theta)dt+g(t)d\bar{w}$  as a MDP by taking $x\_t$ as state, $\pi\_\theta(x\_t,\theta)$ as the selected action, $x\_{t-dt}$ is the next state and the equation 1 itself stands for the transition function.
>
> **Q5: Organization of the paper.**
>
> **"The organization of the paper can be improved to increase clarity."**
>
> **"Figure 1 and 2 needs to be further up in the paper, as they are the motivating examples for authors’ rationale of the paper."**
>
> A5: Thanks for your valuable suggestions. Because time is limited, we have not yet had time to make adjustments. In the next revision, we will advance these two pictures as you suggested.
>
> **References:**
>
>
> [1] Silver, David, et al. Mastering the game of go without human knowledge. Nature (2017).
>
> [2] Haarnoja, Tuomas, et al. Soft actor-critic: Off-policy maximum entropy deep reinforcement learning with a stochastic actor. ICML (2018).
>
>
> [3] Song, Yang, et al. Score-Based Generative Modeling through Stochastic Differential Equations. ICLR (2021).
>
> [4] Anderson, Brian DO. Reverse-time diffusion equation models. Stochastic Processes and their Applications (1982).

---

> ### Author Response · Authors · 2023-11-22
> **Gentle Reminder**
>
> Dear Reviewer QQeh,
>
> Thanks again for your valuable feedback! We sincerely appreciate the time and effort you have dedicated to reviewing our paper.
>
> To respond to your comments, we have made more detailed explanations of our motivation and method, and also provided a more clear and comprehensive background about SDEs.
>
> As the discussion phase will end tomorrow, we kindly request, if possible, that you review our rebuttal at your earliest convenience. If you have any other concerns, we would like to further discuss. If we have addressed your concerns, we sincerely hope you can reconsider the evaluation of our paper.

---

### Official Review · Reviewer_eNtv · 2023-11-01

**Soundness:** 3 good
**Presentation:** 3 good
**Contribution:** 3 good
**Rating:** 6
**Confidence:** 2

**Summary:**

This paper proposes a regularization methods to improve the insufficient coverage issues in policy gradient estimation for the task of applying reinforcement learning to fine-tune diffusion model. Numerical result on a highly-relevant example shows promise of this method comparing to existing a number of baselines.

**Strengths:**

The method has good intuition and is interesting. Numerical performance of the proposed method outperforms the other baselines by a substantial margin.

**Weaknesses:**

It was not clear why reinforcement learning can effectively respect the boundary conditions of the underlying problem while supervised learning approach fail to do so beyond the numerical results. How would an alternative method fail, for example, one that encodes both model matching score and the desired properties into the objective function and then solve the corresponding supervised learning problem, was not articulated.

**Questions:**

- Is the superiority shown in Figure 3 (and also Figure 5 in the appendix) a statistically consistent behavior across majority of examples? Are the numbers shown in the table in Figure 4 possible to have the standard deviation annotations next to the mean and medium values?
- It is surprising that from Figure 4 it seems that TargetDiff does not respect the boundary at all, which seems to be contradicting one of the contributions claimed by the TargetDiff authors. Are there some structural properties associated with these scenarios? Is this a consistent behavior in the majority of the cases?

---

> ### Author Response · Authors · 2023-11-21
> **Response to Reviewer eNtv**
>
> Thanks for your feedback. Please see below for our responses to the comments.
>
> **Q1: Why not directly include desired properties in the objective?**
>
> **"It was not clear why reinforcement learning can effectively respect the boundary conditions of the underlying problem while supervised learning approach fails to do so beyond the numerical results. How would an alternative method fail, for example, one that encodes both model matching score and the desired properties into the objective function and then solves the corresponding supervised learning problem, was not articulated."**
>
> A1: Thanks for your suggestions. We want to clarify that we have already included the desired properties in the objective, as the goal of reinforcement learning is to efficiently and reliably optimize properties through interactions. And policy gradient-based methods are the SOTA of this research field in various applications, please refer to [1] for a comprehensive survey about RL and policy gradients. Therefore, the algorithm that encodes both score-matching and the desired properties as you mentioned, exactly falls into the class of reinforcement learning. Our contribution is a novel policy gradient-estimator for SDE-based policy. With our policy gradient estimator, the optimization process becomes more stable and more efficient.
>
> **Q2: Is the superiority statistically consistent across majority of examples?**
>
> **"Is the superiority shown in Figure 3 (and also Figure 5 in the appendix) a statistically consistent behavior across majority of examples? Are the numbers shown in the table in Figure 4 possible to have the standard deviation annotations next to the mean and medium values?"**
>
> A2:
> Yes. In Figures 5, 6, 7 in Appendix C, we plot the optimization curves of generated ligand molecules over 100 different protein pockets.  As obviously observed, in most cases, our method outperforms all the baselines.
>
> More specifically, our method outperforms TargetDiff in 100 out of 100 cases, outperforms baselines in 75 out of 100 cases, and is at least comparable with the best baseline in almost all cases.
>
> Moreover, as you suggested, we have reported the standard deviation in Table 1 in the revision.
>
>
> **Q3: Performance of baseline TargetDiff.**
>
> **"It is surprising that from Figure 4 it seems that TargetDiff does not respect the boundary at all, which seems to be contradicting one of the contributions claimed by the TargetDiff authors. Are there some structural properties associated with these scenarios? Is this a consistent behavior in the majority of the cases?"**
>
>
> A3: Thank you for your valuable feedback. We want to assure you that the results produced by TargetDiff are both reliable and reproducible. In many protein instances, TargetDiff respects the boundary conditions. However, as depicted in Figure 4, there are some proteins for which TargetDiff is unable to generate high-quality ligands. This could be due to the structural differences between these proteins and those present in the training set. Our results show that we can further significantly improve the pretrained diffusion model by introducing RL.
>
> Besides, notably, violation of the boundary condition is only one of the factors for bad Vina scores (i.e., the proxy for binding affinity, and also the reward in our method). We take the boundary condition as an example for illustration because it is easy to understand even without any background knowledge about biology and chemistry.
>
>
> **References:**
>
> [1] Arulkumaran, Kai, et al. Deep reinforcement learning: A brief survey. IEEE Signal Processing Magazine (2017).

---

> ### Author Response · Authors · 2023-11-22
> **Gentle Reminder**
>
> Dear Reviewer eNtv,
>
> Thank you for the time and effort you have put into evaluating our submission!
>
> In our response, we have further clarified the motivation, novelty, and contribution of our method, and provided more explanations of experimental results.
>
> We kindly remind you that the discussion phase is coming to the end. Please let us know if there are additional concerns we can address for you. If we have properly addressed your concerns, we sincerely hope you can reconsider the evaluation of our submission.

---

> > ### Comment · Reviewer_eNtv · 2023-11-22
> >
> > Thanks for the report on standard deviation. I want to better understand your answer to Q1 that "the algorithm that encodes both score-matching and the desired properties as you mentioned, exactly falls into the class of reinforcement learning". Are you saying that all algorithms with boundary constraints or with the objective of maximizing score exactly falls into the class of reinforcement learning? To me it seems constrained supervised learning approaches can also be of use here. If your argument is because SOTA on image generation using RL has impressive numerical results than supervised learning with boundary constraints, then it would be nice to include exact references beyond the general survey [1] and the textbook you cited in the introduction and confirm that numerical results are the only reason why RL is useful in the applications you consider, similar to your response to Q2 of Reviewer QQeh.

---

> > > ### Author Response · Authors · 2023-11-22
> > > **Response to Reviewer eNtv**
> > >
> > > Thanks for your response.
> > >
> > > What we are claiming is the general-purpose algorithm which aims to maximize a reward through interactions falls into the class of RL. In addition to the survey [1] we have cited, here we directly quote the first sentence of Section 1.1 from the textbook [2]: "Reinforcement learning is learning what to do—how to map situations to actions—so as to maximize a numerical reward signal."  Therefore,  we thought you are proposing the objective of RL by mentioning "the desired properties into the objective function".
> > >
> > > However, we realized that you might be talking about a very specific algorithm about "algorithms with boundary constraints" and "supervised learning with boundary constraints". But to the best of our knowledge, we are not aware of such an algorithm in the research fields of RL, diffusion models, and SBDD. Therefore, in order to make sure we are on the same page, would you mind providing some references of the algorithm you are talking about? Only with specific literature, we can compare our method against them if they are relevant.
> > >
> > > Moreover, we speculate that the "boundary constraint" that you mentioned might be some kind of regularization that penalizes the coordinates of the atoms of a molecule that has some physical conflicts or collisions with the protein. If so, we need to emphasize again that the physical conflicts or collisions are only one of the factors that account for a bad Vina score. The Vina score is complex and far beyond the simple spatial relation. The final goal is to maximize the Vina score instead of only avoiding such physical conflicts, so only considering such a penalty cannot achieve the goal.
> > >
> > > More generally, our goal is to maximize a reward, i.e., a scalar function of a sample and we do not assume any prior knowledge (e.g., boundary constraint is correlated to Vina scores) about this reward in learning. Therefore, our algorithm could also be applied to more general applications. For example, please refer to our response to Reviewer 4LjP's Q1 and Appendix D in the revision for the experiments on text-to-image generation.
> > >
> > >
> > > Besides, we want to clarify that, for structure-based drug design, TargetDiff [3] is the SOTA method, and, for text-to-image generation, Stable Diffusion [4] is the SOTA one. Both are diffusion models (i.e., SDE-based generative models). Since the final goal is to generate samples with a certain desired property, we apply RL to fine-tune the SOTA SDE-based generative models. And our main contribution is a much more stable and efficient policy gradient method dedicated to this scenario.
> > >
> > >
> > > References:
> > >
> > > [1] Arulkumaran, Kai, et al. Deep reinforcement learning: A brief survey. IEEE Signal Processing Magazine (2017).
> > >
> > > [2] Sutton, Richard S., and Andrew G. Barto. Reinforcement learning: An introduction. MIT press (2018).
> > >
> > > [3] Guan, Jiaqi, et al. 3D Equivariant Diffusion for Target-Aware Molecule Generation and Affinity Prediction. ICLR (2023)
> > >
> > > [4] Rombach, Robin, et al. High-resolution image synthesis with latent diffusion models. CVPR (2022).

---

> > > > ### Comment · Reviewer_eNtv · 2023-11-22
> > > >
> > > > Thanks for responding and acknowledging that maximizing an objective function alone does not exhaust all characteristics of RL.
> > > >
> > > > Thanks also for focusing on the boundary constraints. Maximizing a score function with boundary constraints is only one of these considerations when designing score functions without necessarily using RL. A specific reference for realizing such a method can be found in Ganchev, Kuzman, et al. "Posterior regularization for structured latent variable models." The Journal of Machine Learning Research 11 (2010): 2001-2049.
> > > >
> > > > I am happy to increase my score given the clarification of the motivation detailed in the answer to Q1 of Reviewer QQeh.

---

> ### Author Response · Authors · 2023-11-22
> **Response to reviewer eNtv**
>
> Thank you for raising score and providing a reference. We feel the method from this paper may not be easy to directly adapt to the problems we are considering, i.e., SBDD and image generation. It would be interesting to see some future research to further explore in this direction.

---

### Meta-Review · Area_Chair_d2e5 · 2023-12-08

**Metareview:**

The paper considers reinforcement learning (specifically policy gradient methods) for diffusion models.
They propose an alternative to DDPG and REINFORCE, two classical policy gradient algorithms, to deal with the instabilities associated with the high-dimensional problem of diffusion models. The authors are able to leverage the particular structure of the the problem and use certain consistency conditions, which arise in diffusion models, to design a more stable algorithm and obtain a high score on an available benchmark. The ideas are novel although the contribution is not in a mainstream application area of RL.
However, there is limited experimental evaluation. Moreover, there are questions regarding the objective of the paper, and the paper is judged as not being very clear by the reviewers in key aspect (intention, contribution, and some observed phenomena). We recommend the authors to improve the clarify of the submission by addressing the reviewers' concerns.

**Justification For Why Not Higher Score:**

Experiments and motivation could be more thorough

**Justification For Why Not Lower Score:**

N/A

---

### Decision · Program_Chairs · 2024-01-16

Reject